# Simultaneous Prediction Method for Intestinal Absorption and Metabolism Using the Mini-Ussing Chamber System

**DOI:** 10.3390/pharmaceutics15122732

**Published:** 2023-12-05

**Authors:** Satoshi Kondo, Masateru Miyake

**Affiliations:** 1Department of Drug Metabolism and Pharmacokinetics, Nonclinical Research Center, Tokushima Research Institute, Otsuka Pharmaceutical Co., Ltd., 460-10 Kagasuno Kawauchi-cho, Tokushima 771-0192, Japan; kondo.satoshi@otsuka.jp; 2Department of Drug Safety Research, Nonclinical Research Center, Tokushima Research Institute, Otsuka Pharmaceutical Co., Ltd., 460-10 Kagasuno Kawauchi-cho, Tokushima 771-0192, Japan; 3Business Integrity and External Affairs, Otsuka Pharmaceutical Co., Ltd., 2-16-4 Konan, Minato-ku, Tokyo 108-8242, Japan

**Keywords:** intestinal absorption, intestinal metabolism, transport index (TI), prediction, Ussing chamber, metabolites formation index (MFI), gastrointestinal physiological condition

## Abstract

Many evaluation tools for predicting human absorption are well-known for using cultured cell lines such as Caco-2, MDCK, and so on. Since the combinatorial chemistry and high throughput screening system, pharmacological assay, and pharmaceutical profiling assay are mainstays of drug development, PAMPA has been used to evaluate human drug absorption. In addition, cultured cell lines from iPS cells have been attracting attention because they morphologically resemble human intestinal tissues. In this review, we used human intestinal tissues to estimate human intestinal absorption and metabolism. The Ussing chamber uses human intestinal tissues to directly assay a drug candidate’s permeability and determine the electrophysiological parameters such as potential differences (PD), short circuit current (Isc), and resistance (R). Thus, it is an attractive tool for elucidating human intestinal permeability and metabolism. We have presented a novel prediction method for intestinal absorption and metabolism by utilizing a mini-Ussing chamber using human intestinal tissues and animal intestinal tissues, based on the transport index (TI). The TI value was calculated by taking the change in drug concentrations on the apical side due to precipitation and the total amounts accumulated in the tissue (T^corr^) and transported to the basal side (X^corr^). The drug absorbability in rank order, as well as the fraction of dose absorbed (Fa) in humans, was predicted, and the intestinal metabolism of dogs and rats was also predicted, although it was not quantitative. However, the metabolites formation index (MFI) values, which are included in the TI values, can predict the evaluation of intestinal metabolism and absorption by using ketoconazole. Therefore, the mini-Ussing chamber, equipped with human and animal intestinal tissues, would be an ultimate method to predict intestinal absorption and metabolism simultaneously.

## 1. Introduction

Drug development has been dramatically changing in recent years due to the development of new analytical methods and synthesis approaches. Thus, antibody conjugates [1,2,3], antibodies [4,5,6], and peptide mimetic compounds [7,8,9,10,11,12,13,14], which are the so-called middle-molecular drug candidates, have been appearing on the market in the past decade. The patients suffering from chronic disease need frequent administration of their drugs. In that case, they must desire to use a painless and convenient administration method in order to continue. Oral administration is the most convenient way to ensure patients’ adherence to their medication because it is non-invasive and painless. Therefore, the intestinal absorption properties of compounds in humans are a critical factor in the early stages of drug development in order to identify oral drug candidates with sufficient bioavailability.

The prediction tool for oral absorbability in humans has reported that using Caco-2 cells, which are the most famous cultured cell lines for judging human absorbability [15,16], might be difficult for drugs with absorbability under 50% because the slope of the estimated theoretical curve is sharp [15] compared with the results of the Ussing chamber study [17]. In the case of using human intestinal tissues in ex vivo studies, we need to use a Ussing chamber system in order to mount the tissues directly [18,19,20,21,22,23,24]. This system was originally developed in 1951 [25] and has been improved through using intestinal tissues in rats, rabbits, dogs, monkeys [26,27,28], and humans [17,20,29,30]. We investigated the comprehensive evaluation between animal and human tissues and correctly reported that the small intestinal tissues from rats and dogs are accurate in predicting human permeability [28]. Utilizing the Ussing chamber system and attaching freshly isolated human jejunum has also been reported to be useful for evaluating the function of intestinal uptake/efflux transporters in intestinal absorption in humans [31]. Furthermore, Michiba et al. have reported human jejunal spheroid from intestinal epithelial cells in humans as a novel in vitro model [32]. They seem to have been successfully established from surgical human jejunal specimens as three-dimensional human intestinal spheroids and expanded for a long time using L-WRN-conditioned medium [32]. Accordingly, the usage of human specimens is attractive for not needing to consider the difference between species.

For the evaluation of intestinal metabolism, transport index (TI) values are able to estimate the human intestinal absorption and metabolism simultaneously using species differences between rats and dogs [33]. In addition, judgement of compound differences would be possible by using TI values with the metabolites formation index (MFI), which is defined as the sum of the percentage of the dose of the produced metabolites corrected using the AUC value of the parent drug in the apical compartment based on the mass balance of the tested compound [34]. The larger the size of human intestinal tissues, the closer the contribution of the intestinal metabolism to the intestinal absorption in order to effect the real enzymatic reaction. We have been attempting to clarify that the tissues, which can be mounted on the 5 mm diameter mini-Ussing chamber, would be useful to estimate the simultaneous quantity of intestinal absorption and metabolism for midazolam and nifedipine, although usage of the 9 mm diameter mini-Using chamber needs further investigation. Furthermore, we have been investigating the information on drug absorption and electrophysiological parameters in the large intestine, such as potential differences (PD), resistance (R), and so on, in humans affected with ulcerative colitis (UC) and Crohn’s disease (CD) [35,36]. 

In this review, we focused on the prediction of human drug absorption by utilizing human intestinal tissues and TI values produced by various techniques of intestinal drug absorption prediction. We also focused on whether the simultaneous prediction of human intestinal absorption and metabolism is possible. Furthermore, we investigated the physiological gastrointestinal conditions such as pH, buffering ability, osmolality, and so on between young adults and older adults. 

## 2. Gastrointestinal Fluid Volumes and Composition

### 2.1. Intestinal Fluid Volumes

The volume of intestinal fluids in the gastrointestinal tract changes depending on the liquid orally ingested from beverages and food, the excretion of water from the bladder, the secretion of pancreatic juices, and the secretion and absorption of intestinal fluids by the gastrointestinal tract. The entire intestinal fluid volume in the gastrointestinal tract determines the volume of drug solubilization and is a key factor in the oral absorption of drugs. Manchikanti et al. reported that the volumes of fluid in the proximal colon, distal ileum, and stomach are affected by age difference [37]. The volume of gastric fluids aspirated from older and young adults scheduled for elective surgery were also determined while in the fasted state. In a comparison across age groups, although the differences among them were not statistically significant, it was found that the mean residual volume of gastric intestinal fluids decreases in older adults. Participants included non-obese patients scheduled for surgery with no gastrointestinal disorders and not receiving H_2_-inhibitors, anticholinergics or gastrokinetic agents. Before the surgery, some of the participants were medicated. It should be noted that the data may have been influenced by comorbidities. However, it appeared that an age-dependent relationship existed in terms of the fluid volumes collected from the proximal colon and the distal ileum. Both 5 h after meal intake and in the fasted state, the intestinal volumes in young adults were significantly greater than those in older adults [37]. The thicker mucus layer in older adults, which is difficult to collect, may be at least partly responsible for this difference. 

### 2.2. Gastrointestinal pH, Buffering Ability and Osmolality

In the fasted state, the pH of gastrointestinal fluids is normally lower than fluids in other parts of the gastrointestinal tract because the proton pump secretes gastric acid that is required for the digestion of proteins. The reference data regarding pH values in gastric fluids in young and older adults are summarized in Table 1 [38]. The data from young and older adults on pH values in gastric fluids are summarized in the reference section [39]. In both the fasted state (water intake allowed) and in the fed state, it should be noted that these conditions are not at all static. There is virtually no such dynamic data for older adults. Using the Heidelberg capsule developed by Dressman et al., it was found that the gastric pH in older adults is significantly lower than in young adults in the fasted state [40]. 

In this study, participants were asked to discontinue smoking 3 days prior to measurement. In contrast, Pedersen et al. enrolled non-smokers without any gastrointestinal diseases, but they were not allowed to drink water in the 2 h before the aspiration of gastric contents [39]. 

In this study, a tendency for gastric pH to be lower was observed in younger adults when compared to older adults. In another study conducted in the fed state using the Heidelberg capsule, the results indicated that the gastric pH in both young and older adults was almost the same, but the time taken for stomach fluids to return to pH 2 was significantly shorter in younger adults compared to older adults after meal intake [37,41]. 

The impact of older adult age on acid production has been investigated in gastroduodenal diseases as well. Feldman et al. have reported that their study included individuals without pernicious anemia or malignancies, and that were not receiving medications that could have had an impact on gastric secretion [42]. They also evaluated when peak acid production occurred [42]. They showed that acid production decreases with age, and multivariate analysis revealed an age-dependent effect on acid production [42]. It was found that a lower prevalence of smoking and higher prevalence of atrophic gastritis in older adults explained the decline in acid production. Based on these data, the impact of H. pylori colonization on gastric intestinal acid production as a function of age remains unclear. Katelaris et al. have reported that H. pylori infection has no effect on basal or peak acid production in both young adults and older adults [43].

It has also been reported that the existence of H. pylori in biopsy samples was associated with increased peak acid production [43]. The pH values in the duodenum, measured using the Heidelberg capsule, were significantly lower in young adults compared with older adults [40,41]. The reference data regarding pH values in the duodenum, distal ileum and proximal colon fluids have been determined in young and older adults and are summarized in Table 2 [38]. In contrast, the pH measured in intestinal contents collected from volunteers, of which there were a limited number, was almost the same between older adults and young adults in a study conducted by Annaert et al. [44]. A similar observation was made for intestinal fluids collected from the distal ileum during colonoscopy, where the pH was not significantly different between young adults and older adults [44]. However, the pH of intestinal fluids collected from the proximal colon was significantly higher in young adults [45]. The pH measured with the Heidelberg capsule in the duodenum was significantly lower in young adults compared to older adults in the fed state [40,41]. 

For older adults, valuable data exist regarding the impact on the buffer ability of luminal fluids or contents of the small intestine and the stomach [45,46,47]. The reference data for the buffer capacity of fluids in the distal ileum and proximal colon have been determined in young and older adults and are summarized in Table 3. Some of the age-related differences were observed in the distal segments of the intestinal lumen. The luminal contents in the proximal colon were shown to exhibit a significantly lower buffer ability in young adults compared to older adults in the fasted state. Similarly, the buffer ability in the distal ileum contents was lower in young adults than the mean value in older adults at 5 h after meal intake. In accordance with the data mentioned above, the proximal colon samples collected from young adults at 5 h after meal intake showed lower buffer ability compared to older adults in Table 3.

The impact of older adult age on the osmolality of gastric fluids has yet to be investigated, while some data on the osmolality of intestinal contents are available. The reference data for the osmolality of contents of the duodenum, distal ileum and proximal colon have been determined in young and older adults and are summarized in Table 4. The osmolality of aspirated duodenal contents from young adults is not significantly different from older adults in the fasted state. However, the values in the proximal colon and the distal ileum of older adults were more than twice those of young adults [44,45,46,47]. While this difference was especially noticeable in the fasted state, the osmolality values were comparable again between age groups at 5 h after meal intake [44,45,46,47]. In summary, based on the information mentioned above, little is known about the impact of older age on the osmolality, buffer capacity, and intraluminal pH of luminal contents. Further information is needed as we do not have information on the potential impact of systemic diseases on the osmolality, pH, or buffer ability of gastrointestinal fluids in individuals of advanced age.

## 3. Intestinal Transporters and Epithelial Metabolism Enzymes

The intestinal epithelium possesses numerous functions such as absorption, excretion, and metabolism, as well as serving as a barrier to foreign substances and materials. It can uptake essential elements like nutrients through transporters that mediate cellular absorption [48].

The main character of the intestinal epithelium, in which transporters and metabolic enzymes are expressed, is based on passive diffusion, augmented by the segment-dependent, non-homogenous expression of various transporters and metabolic enzymes along the intestinal tract [48]. Transporters control uptake or efflux as gate keepers and are vital in understanding the permeation behavior of drugs used in clinical practice. Intestinal transporters belong to two major superfamilies, the solute carrier transporters (SLC) and the ATP-binding cassette transporters (ABC), which presently include over 400 and 49 transporters, respectively [49]. For instance, in Figure 1, the protein concentrations and subcellular localizations of transporters in a static Caco-2 culture system are presented [50]. In addition, a mean-centered sigma-normalized heatmap of transcriptomic differences in various in vitro models, as well as in the human jejunum, ileum and duodenum, is presented in Figure 2 [50]. The recent proteomics research has provided a quantitative comparison of ABC or SLC drug transporters and intestinal enzymes between Caco-2 cells, typical cell culture lines used in drug absorption studies, and human tissues. It seems that the result of this research reveals differences between Caco-2 cells and human tissues [50].

In spite of the hundreds of known transmembrane proteins such as cytochrome P450 (CYP), the knowledge of SLC and ATP transporter abundance, function, and localization in the human intestine is limited, but this knowledge is now beginning to expand. The current understanding is described well by Müller et al. [48]. The uptake transporters in the intestine like PEPT1 (human peptide transporter 1), MCT1 (monocarboxylate transporter 1), and LAT2 (L-type amino acid transporter 2) are able to accelerate the uptake of peptide-like structures (ß-lactam antibiotics, ACE inhibitors, etc.), monocarboxylates (valproic acid etc.), amino acid-like compounds (L-Dopa, melphalan etc.), and prodrugs of antiviral compounds (valacyclovir etc.) [51,52].

Intestinal efflux transporters, which are P-gp (P-glycoprotein), MRP2 (multidrug resistance protein 2), and BCRP (breast cancer resistance protein), belong to the ABC transporter superfamily. These are expressed in the enterocytes on the apical membrane, and promote transport into the gut lumen from the enterocytes. Therefore, they could potentially limit the uptake of drugs such as antibiotics, immunosuppressants, cytostatic drugs, and statins [48]. 

Xenobiotics are detoxified and/or bioactivated by a combination of both phase I and phase II metabolism across the intestinal epithelium. Phase I metabolism is mainly conducted by oxidation, hydrolysis reactions, and reduction. In particular, oxidative reactions catalyzed by the members of CYP superfamily are the most common in the reference [53]. In Table 5, typical CYP enzymes and the major drug transporting ABC and SLC proteins in the human jejunum are listed [54].

Sulfotransferase and UDP-glucuronosyltransferase families, which conjugate xenobiotics with endogenous ligands and make more water-soluble compounds in order to detoxify, are phase II enzymes in the intestine [53]. Although the processes of biological change in the intestinal epithelium have not been investigated in the previous decades, Peters et al. have reported that they are a key factor in the absorption process, at least in marketed drugs such as midazolam, cyclosporin, statins, and docetaxel [55].

As the intestine is important in the process of the absorption and metabolism of, at least, marketed drugs, understanding the age effect on intestinal enzymes and transporter expression is of major importance. The effect of disease-independent aging will be discussed in the following. Lindell et al. investigated the mRNA expression of CYP1A1, CYP1A2, CYP2A6, CYP2B6, CYP2C9, CYP2D6, CYP2E1, CYP3A4, and P-gp in the duodenal biopsies from 51 adults undergoing gastroscopy [56]. 

Correlations among various indexes were assessed, including age, diet, smoking and drug usage. It was not observed that the age effect was significance on the expression of CYP450 or P-gp [56]. Lown et al. investigated that the CYP3A4 and P-gp protein expression by Western blot analysis was abundant even the duodenum of kidney transplanted patients. It was found that expression of intestinal P-gp and CYP3A4 was not changed dynamically by age [57]. 

Larsen et al. investigated the activity of intestinal P-gp by referring to the area under the curve of plasma concentration in the first 4 h after ingestion of digoxin as a representative parameter that reflects intestinal P-gp activity. The changes that were observed in its activity were independent of age [58]. 

Miki et al. found that the tissues obtained from autopsy did not show any significant effect of age on the intestinal P-gp and CYP3A4 mRNA expression. However, a tendency towards lower P-gp and CYP3A4 expression in the small and large intestines was seen in the mRNA level with increasing age [59].

Recently, based on the limited data available, it was not observed that an age-dependent effect on the expression of drug metabolic enzymes and transporters is apparent. However, the valuable studies are mainly based on covariate analysis of data from clinical trials, which include subjects with comorbidities or whose health status is unclear. Two recent studies using the newest quantitative liquid chromatography with tandem mass spectrometry (LC-MS/MS)-based proteomics are in progress to investigate a variety of ABC-transporters, SLC-transporters, CYP450s, and UGTs. Neither of the two studies reported an age-dependent effect on protein expression, but one study included pathologic obese patients and the other included subjects with heterogeneous health statuses, including those with ischemia, inflammatory bowel disease, and colon cancer [60,61]. 

As there is increasing evidence that the expression volume of enzyme and transporter could be affected by certain conditions, the enrollment of patients suffering from inflammatory bowel disease, obesity, and kidney failure needs to be avoided [62]. Apart from the deficiency caused by the enrollment of diseased patients, the selection of an appropriate quantification method is the most important factor. The quantification of mRNA as a representative for abundance is limited and should be interpreted with caution because the intestine is one of the high turnover organs [60]. On the other hand, protein abundance determined using Western blot assay often has the problems of limited reproducibility and unknown accuracy and precision [63]. The advent of LC-MS/MS-based targeted proteomics represents a solution by providing good selectivity, precision, and accuracy with rapid analysis times [64].

In spite of the aforementioned advantages, the amount of targeted proteomics data regarding the effect of aging on drug metabolic transporter and enzyme expression is still small. Therefore, a clear need still exists for a new and highly advanced method for quantifying protein abundance in healthy older volunteers so that a definitive determination on the impact of age on intestinal drug transporter expression and metabolic enzymes can be made.

## 4. The Simultaneous Prediction of Human Intestinal Absorption and Metabolism Using the Transport Index (TI)

### 4.1. The Calculation of the Transport Index (TI)

The usage of TI values for predicting intestinal drug absorption is expected to be a powerful tool because it can evaluate human intestinal absorption more correctly than other systems, such as Caco-2 shown in Figure 3 and Figure 4, which perform evaluations with P_app_ values, and it has strong validity and reproducibility for human intestinal absorption with Fa in humans in rank order [29]. To understand more easily, we show a schematic illustration in Figure 5 with the explanation and calculation method of TI values following below.

To estimate the permeability based on the actual drug concentration in the apical compartment, we defined the transport index (TI), the percentage of dose transported into the basal side from the apical side, and the tissue-accumulated drug concentration corrected using the AUC value of drug in the apical compartment (Equation (1)).
(1)$$TI=X^{corr}+T^{corr}$$
where X^corr^ and T^corr^ are the percentage of dose transported into the basal side compartment and of dose accumulated in the tissue, respectively, which were corrected using the AUC value of drug concentration in the apical compartment (Equations (2) and (3)).
(2)$$\begin{array}{l} X^{corr}=[\frac{2\cdot \left(C_{apical}^{t_{0}}\cdot t_{3}\right)\cdot \left(C_{basal}^{t_{3}}\cdot V_{basal}\right)}{\sum_{i=1}^{3}((C_{apical}^{t_{i-1}}+C_{apical}^{t_{i}})\cdot (t_{i}-t_{i-1}))}\\ \;\;\;\;\;+\sum_{i=4}^{6}\left(\frac{2\cdot (C_{apical}^{t_{0}}\cdot (t_{6}-t_{3}))\cdot (C_{basal}^{t_{i}}\cdot V_{basal}-C_{basal}^{t_{i-1}}\cdot (V_{basal}-V_{sampled})}{\left(C_{apical}^{t_{i-1}}+C_{apical}^{t_{i}})\cdot (t_{i}-t_{i-1}\right)}\right)]\\ \;\;\;\;\;\cdot \frac{1}{C_{apical}^{t_{0}}\cdot V_{apical}}\cdot 100\;(\%) \end{array}$$
(3)$$T^{corr}=\frac{2\cdot C_{apical}^{t_{0}}\cdot t_{6}\cdot T}{\sum_{i=1}^{6}\left(C_{apical}^{t_{i-1}}+C_{apical}^{t_{i}})\cdot (t_{i}-t_{i-1}\right)}\cdot \frac{1}{C_{apical}^{t_{0}}\cdot V_{apical}}\cdot 100\;(\%)$$
where $C_{apical}^{t_{i}}$ and $C_{basal}^{t_{i}}$ mean the drug concentrations in the apical side and basolateral side at time t_i_, respectively. $V_{apical}$, $V_{basal}$, and $V_{sampled}$ represent the volume of the apical compartment (1.35 mL), the volume of the basal compartment (1.35 mL), and the sampled volume from basal compartment (0.6 mL), respectively. T is the amount of drug accumulated in the intestinal tissue at the end of the transport experimental study. Furthermore, t_0_ to t_6_ mean 0, 0.083, 0.25, 0.5, 1, 1.5, and 2 h, respectively.

In addition, alternative methods utilizing TI values have already been developed by using rats and dogs [28].

### 4.2. The Simultaneous Prediction of Intestinal Absorption and Metabolism Using the Transport Index (TI)

Simultaneous prediction has been carried out by using small intestinal tissues from rats and dogs [33]. Midazolam, a highly permeable and highly soluble drug that is categorized as class I in the Biopharmaceutical Classification System (BCS), was used as a model compound and a typical CYP3A4 substrate. The TI value of midazolam was 7.188 in rats. Due to this value, midazolam was recognized as a well-absorbed drug, hence its categorization as a BCS class I drug. The TI value of midazolam significantly increased to 8.361 in the presence of ketoconazole (Figure 6A). Then, metabolite formations of 1-hydroxymidazolam (1-OH) and 4-hydroxymidazolam (4-OH) in rats were detected, and the TI value of 1-OH was higher than that of 4-OH. Thus, ketoconazole significantly inhibited the formation of 4-OH (Figure 6B). Moreover, ketoconazole decreased the tissue accumulation of 1-OH and 4-OH in rats [33]. In the case of dogs, the TI value of midazolam was 4.299, which was lower than in rats. The addition of ketoconazole in dogs showed a trend to increase the TI value of midazolam; however, it was not statistically significant (Figure 7A). Both 1-OH and 4-OH were detected as metabolites in dogs, and the TI value of 4-OH was higher than that of 1-OH. The tissue accumulation of 1-OH and 4-OH seemed to be decreased in dogs (Figure 7B).

The metabolite formation of 4-OH was lower than 1-OH in rats (Figure 6B), which made it possible to note that the intrinsic clearances of 1-OH were larger than those of 4-OH in rat intestinal microsomes [62]. The difference in the order of metabolite formations could be due to substrate inhibition kinetics, which were observed in dog liver microsomes [65]. In addition, it was reported that the maximum velocity (Vmax) values for the formations of 1-OH and 4-OH showed a difference of two-fold or less in dog liver microsomes [63]. Further discussion is needed on the differences in kinetic analysis. Although significant differences were not observed between the control and ketoconazole group in rats, except in the TI of 4-OH (Figure 6B), the TI values of 1-OH and 4-OH in rats and dogs showed a tendency for inhibitory effects with ketoconazole. Kotegawa et al., have already reported that ketoconazole inhibits the formation of 1-OH and 4-OH in rat liver and intestinal microsomes [66]. In particular, it seems that the inhibitory effect on 4-OH formation in dog liver microsomes was also observed [67]. Accordingly, in this mini-Ussing chamber system, the inhibitory effects of ketoconazole on the metabolite formation of midazolam were similar to the observations reported in the literature [68,69,70].

However, the metabolic capacity of the mini-Ussing chamber system is limited, although there are at least three sizes of intestinal tissue, 3, 5, and 9 mm. The ratio of midazolam metabolites to unchanged midazolam calculated using the TI values was 0.57% and 0.51% in rats and dogs, respectively [33]. The metabolic activity in intestinal tissues utilized by the Ussing chamber system was low when compared to that of in vitro studies using microsomes [66]. As healthy intestinal tissues are able to have an enzyme source, larger intestinal tissues would have the potential to increase the amount of metabolites because the enzyme source becomes large when compared to smaller intestinal tissues. Furthermore, the enzyme activity of these tissues would decrease with tissue edge damage [71]. With regard to the edge effect, it has been reported that the modified Ussing chamber system would offer an easy way to handle small biopsy samples with a minimal degree of edge effect [72]. Accordingly, as a next step, we intended to expand the hole diameter and improve the intestinal enzyme source in order to minimize the edge effect of intestinal tissue.

## 5. The Simultaneous Prediction of Intestinal Absorption and Metabolism Using Metabolites Formation Index (MFI)

### 5.1. The Calculation of Metabolites Formation Index (MFI)

It seems that TI values could also potentially evaluate intestinal metabolism [33]. However, as the metabolic activity in human intestinal tissues was lower when compared to that of in vitro studies [64], the larger hole diameter of the chamber was used because of the high enzyme activity and decreasing edge tissue damage. Furthermore, we attempted to clarify the contribution of intestinal metabolism to the intestinal availability [34].

To estimate the metabolites formation based on the parent concentration in the apical compartment in the mini-Ussing chamber system, metabolites formation index (MFI) values were used. MFI value was defined to be the sum of the percentage of the dose of the produced metabolite, corrected using the AUC value of the parent drug, in the apical compartment based on the mass balance of tested compound (Equation (4)).
MFI = X^corr^+ T^corr^+ E^corr^(4)
where X^corr^, T^corr^, and E^corr^ are the percentages of the dose appearing as a metabolite to the apical compartment, appearing as a metabolite to the basal compartment, and accumulated in the tissue, respectively, which were corrected using the AUC of test compounds in the apical compartment (Equations (5)–(7)).
(5)$$\begin{array}{l} X^{corr}=[\frac{2\cdot (C_{apical}^{t_{0}}\cdot t_{3})\cdot (C_{basal}^{t_{3}}\cdot V_{basal})}{\sum_{i=1}^{3}((C_{apical}^{t_{i-1}}+C_{apical}^{t_{i}})\cdot (t_{i}-t_{i-1}))}\\ \;\;\;\;\;+\sum_{i=4}^{6}\left(\frac{2\cdot (C_{apical}^{t_{0}}\cdot (t_{6}-t_{3}))\cdot (C_{basal}^{t_{i}}\cdot V_{basal}-C_{basal}^{t_{i-1}}\cdot (V_{basal}-V_{sampled})}{\left(C_{apical}^{t_{i-1}}+C_{apical}^{t_{i}})\cdot (t_{i}-t_{i-1}\right)}\right)]\\ \;\;\;\;\;\cdot \frac{1}{C_{apical}^{t_{0}}\cdot V_{apical}}\cdot 100\;(\%) \end{array}$$
(6)$$\begin{array}{l} E^{corr}=[\frac{2\cdot (C_{apical}^{t_{0}}\cdot t_{3})\cdot (C_{basal}^{t_{3}}\cdot V_{basal})}{\sum_{i=1}^{3}((C_{apical}^{t_{i-1}}+C_{apical}^{t_{i}})\cdot (t_{i}-t_{i-1}))}\\ \;\;\;\;\;+\sum_{i=4}^{6}\left(\frac{2\cdot (C_{apical}^{t_{0}}\cdot (t_{6}-t_{3}))\cdot (C_{basal}^{t_{i}}\cdot V_{basal}-C_{basal}^{t_{i-1}}\cdot (V_{basal}-V_{sampled})}{\left(C_{apical}^{t_{i-1}}+C_{apical}^{t_{i}})\cdot (t_{i}-t_{i-1}\right)}\right)]\\ \cdot \frac{1}{C_{apical}^{t_{0}}\cdot V_{apical}}\cdot 100\;(\%) \end{array}$$
(7)$$T^{corr}=\frac{2\cdot C_{apical}^{t_{0}}\cdot t_{6}\cdot T}{\sum_{i=1}^{6}\left(C_{apical}^{t_{i-1}}+C_{apical}^{t_{i}})\cdot (t_{i}-t_{i-1}\right)}\cdot \frac{1}{C_{apical}^{t_{0}}\cdot V_{apical}}\cdot 100\;(\%)$$
where $C_{apical}^{t_{i}}$, $C_{intactapical}^{t_{i}}$, and $C_{basal}^{t_{i}}$ represent the drug concentrations in the apical side and basal side at time t_i_, respectively. $V_{apical}$, $V_{basal}$, and $V_{sampled}$ represent the volume of the apical compartment (1.35 mL), volume of the basal compartment (1.35 mL), and sampled volume from the basal compartment (0.6 mL), respectively. T is the amount of the drug accumulated in the intestinal tissues t the end of the transport experimental study. Furthermore, t_0_ to t_6_ mean 0, 0.083, 0.25, 0.5, 1, 1.5 and 2 h, respectively.

### 5.2. The Simultaneous Prediction of Intestinal Absorption and Metabolism Using the Metabolites Formation Index (MFI)

The TI value of midazolam was 12.04 and in the presence of ketoconazole, it significantly increased to 13.40 (Figure 8A). The ratio of its metabolites, such as 1-OH and 4-OH, to the absorbed sum was 5.47%. In the presence of ketoconazole, the MFI value for midazolam significantly decreased from 0.70 to 0.02 (Figure 8B). The TI value of nifedipine was 8.56 in rats, which was lower than that of midazolam (Figure 8A and Figure 9A). In the presence of ketoconazole, the TI value of nifedipine in rats significantly increased to 10.79 (Figure 9A). The ratio of its metabolite, oxidized nifedipine, to the absorbed sum was 13.42%. In the presence of ketoconazole, the sum value of MFI on nifedipine significantly decreased from 1.58 to 1.12 (Figure 9B). The simultaneous prediction of both intestinal absorption and metabolism was determined again, utilizing small intestinal tissues from rats, because we were able to choose between rats and dogs from the rest of animal species [28,33]. Midazolam, categorized as class I in the BCS, was used as a typical model compound [72,73]. Nifedipine, a drug with high permeability and low solubility in BCS class II [74], was used as a model compound and a well-known CYP3A4 substrate. The TI ratios of midazolam and nifedipine with the addition of ketoconazole, which is a CYP3A4 inhibitor, were 1.11 and 1.26, respectively (Figure 8A and Figure 9A), suggesting that intestinal metabolism causes a decrease in the intestinal intrinsic clearance of both compounds. Furthermore, the sum value of TI was significantly improved for nifedipine in the presence of ketoconazole [34], which demonstrates an inhibition of the efflux transport of nifedipine via P-gp or an increase in nifedipine uptake. Dorababu et al. have reported that nifedipine is a typical P-gp substrate [75]. Thus, an apparent Fa of nifedipine could be increased by the inhibitory effect via intestinal P-gp. Regarding the contribution of metabolizing enzymes and P-gp in intestinal tissues, further investigation on nifedipine disposition is needed. In addition, the TI ratio of nifedipine suggested that it is the easier compound for intestinal metabolism compared to midazolam. To be precise, the intestinal intrinsic clearance (Fa·Fg) of nifedipine and midazolam in rats has been reported to be 0.579 for nifedipine and 0.771 for midazolam [76], even though the absolute bioavailability value of nifedipine and midazolam in rats was from 0.462 and 0.011 to 0.045, respectively, after oral administration [37,77,78]. With regard to intestinal metabolism, the Fa·Fg value of nifedipine was about 1.4–fold larger than that of the control. On the other hand, in the case of midazolam, it was 31–fold larger than the control due to the inhibitory effect of ketoconazole on intestinal CYP3A activities (Figure 8B and Figure 9B). Taken all together, it was suggested that our system would make it clearer to evaluate the contribution of intestinal metabolism to intestinal absorption. On the other hand, further evaluation using compounds known for intestinal bioavailability is needs to be reproducible for other compounds as well.

The potential advantage of the mini-Ussing chamber system compared to the conventional chamber system is that it would minimize the adsorption of tested compounds due to its smaller size, especially the volume on both the donor and receiver sides. However, the edge tissue damage on the 3 mm diameter mini-Ussing chamber could change the capacity of intestinal metabolism because the ratio of midazolam metabolites, 1-OH and 4-OH, to unchanged midazolam calculated using TI values was only 0.57% [33,67,68]. Therefore, we used the mini-Ussing chamber with a 5 mm hole diameter as a device. The exposed area on the 5 mm diameter increased to 2.8–fold larger than that of the 3 mm diameter. The ratio of midazolam metabolites to unchanged midazolam was 5.47% calculated with TI value. Thus, it was about 10-fold larger than that of the 3 mm diameter [33], suggesting that the edge tissue damage was improved disproportionally from the viewpoint of the exposed area 5 mm in diameter. As described above, we observed that the edge effect on the 3 mm diameter was larger than on the 5 mm diameter, considering the improvement of the edge tissue damage on the 5 mm diameter. In addition, the rank order of intestinal availability was well correlated between nifedipine and midazolam due to improvement in the enzyme source of intestinal tissues. Furthermore, the TI value of midazolam in rats was higher than that obtained in the previous article with the 3 mm diameter chamber [33].

## 6. Conclusions

The mini-Ussing chamber system, which is equipped with human and animal intestinal tissues, showed the potential for simultaneous evaluation of intestinal absorption and metabolism by utilizing TI and MFI indexes, which were novel parameters. In addition, the TI values in this system could detect species differences, and the MFI index could detect compound differences in intestinal metabolism as well. As the Ussing chamber system is the only method to use human tissue directly as an in vitro system, we would find information more accurately than in other in vitro systems. Furthermore, the morphological gastrointestinal tract conditions, such as gastrointestinal pH, buffering ability, and osmolality, for areas including the stomach, duodenum, ileum and colon could be important to know in order to understand how foods are digested and absorbed, with attention to the effect of aging as well.

## Figures and Tables

**Figure 1 pharmaceutics-15-02732-f001:**
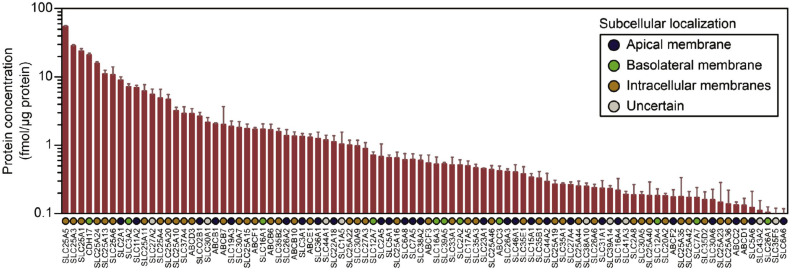
Protein concentrations and subcellular localizations of transporters that were present at concentrations > 0.1 fmol/mg protein in static Caco-2 transwell cultures after 21 days of culture. Figure received with permission from [50].

**Figure 2 pharmaceutics-15-02732-f002:**
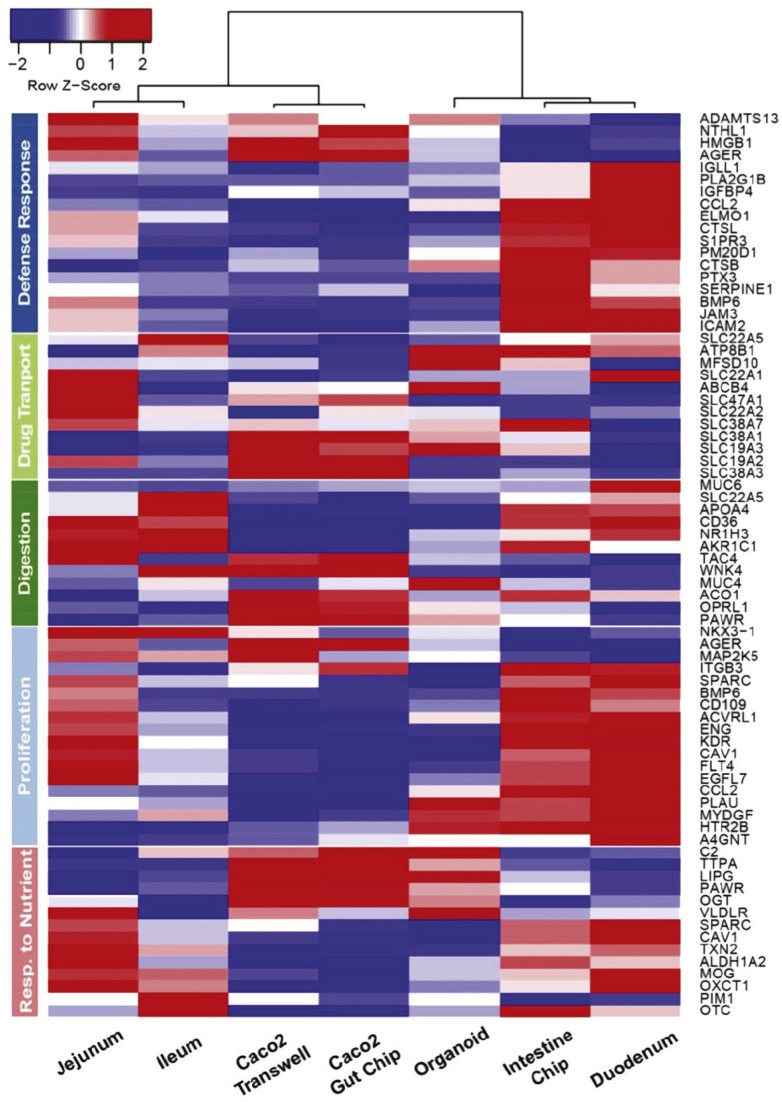
Mean-centered sigma-normalized heatmap representation of transcriptomic differences in various in vitro models, as well as human jejunum, ileum and duodenum. Note that 3D organoids and the microfluidic chip refer to the same duodenal tissue biopsies derived from three healthy donors. Figure received with permission from [50].

**Figure 3 pharmaceutics-15-02732-f003:**
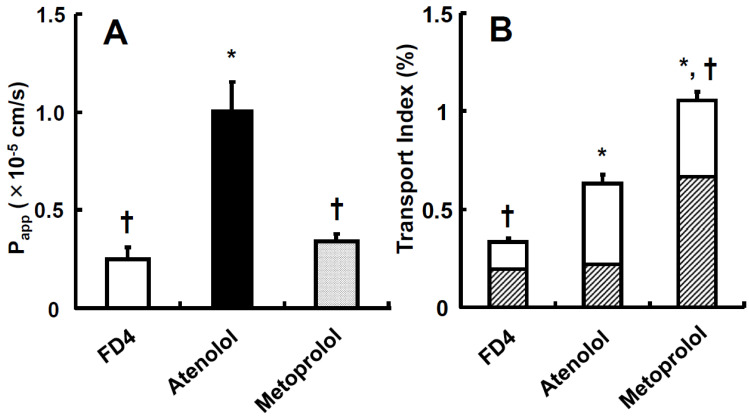
Apparent permeability (P_app_) and Transport index (TI) of FD-4, atenolol, and metoprolol across human intestinal tissues. (**A**) Human small intestinal tissues (ileum) from CD patients were mounted in a mini-Ussing chamber. P_app_ was calculated by the standard equation which follows below. P_app_ = dQ/dt ·1/(A · C0). (**B**) TI, X^corr^ and T^corr^ were calculated using Equations (1)–(3). Each value of TI is the mean with SE of twelve experiments. * *p* < 0.05 compared with FD-4, † *p* < 0.05 compared with atenolol. Open column for X^corr^; closed column for T^corr^. Figure with permission from [29].

**Figure 4 pharmaceutics-15-02732-f004:**
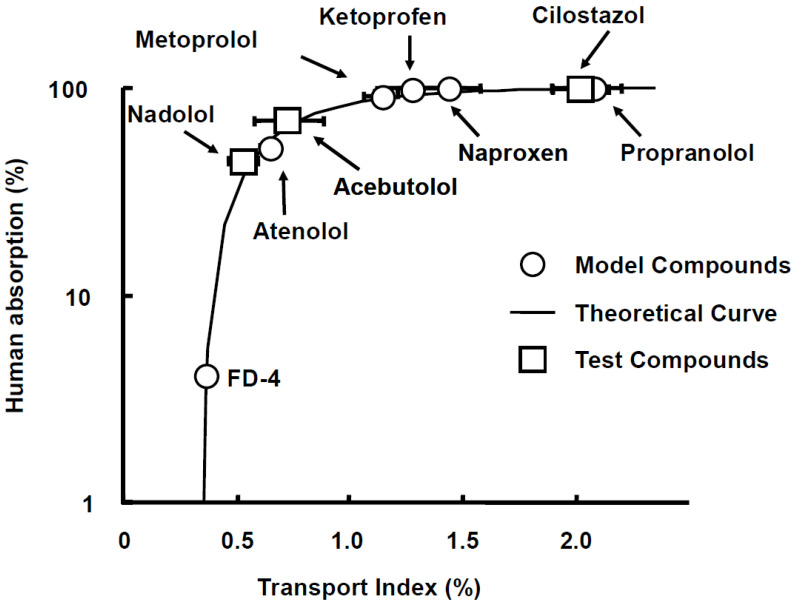
Relationship between transport index (TI) and fraction of dose absorbed (Fa) in humans for drugs with different characteristics. Fa values of drugs cited from references were plotted against TI values. Theoretical fitting line was obtained for six model compounds calculated using $\mathrm{Fa}\;\mathrm{in}\;\mathrm{humans}(\%)\;=\;100\cdot (1-e^{-f\cdot (\mathrm{TI}-\alpha )})$ from [20]. The value of f and *α* were obtained as 2.347 ± 0.230 and 0.314 ± 0.002, respectively, from [20]. The figure was received with permission from [29].

**Figure 5 pharmaceutics-15-02732-f005:**
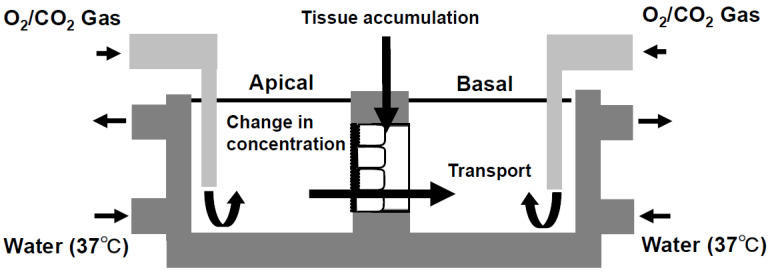
Schematic illustration of a mini-Ussing chamber system utilizing human intestinal tissue. Human intestinal tissues were mounted between the apical and basolateral chambers. Both sides were filled with transport buffer (apical and basolateral side; pH = 7.2, volume = 1.35 mL). Figure received with permission from [29].

**Figure 6 pharmaceutics-15-02732-f006:**
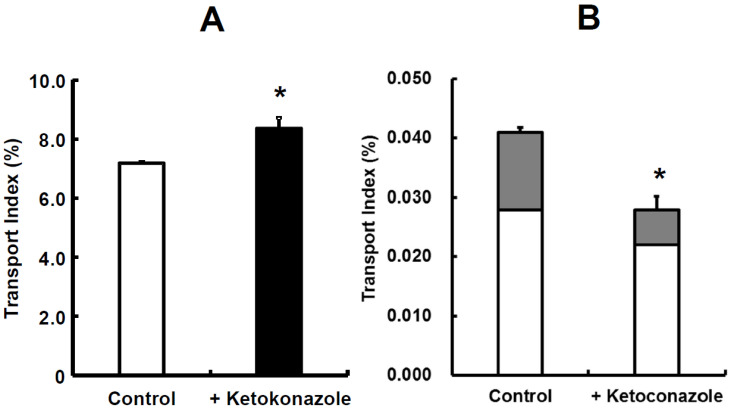
The inhibitory effect of ketoconazole on absorption of midazolam (**A**) and its metabolites (**B**) in rat small intestinal tissues. Each value of TI is the mean with SE of three experiments. * *p* < 0.05 compared with data in the absence of ketoconazole. In Figure 3B, the open column denotes 1-OH; closed column denotes for 4-OH. Figure received with permission from [33].

**Figure 7 pharmaceutics-15-02732-f007:**
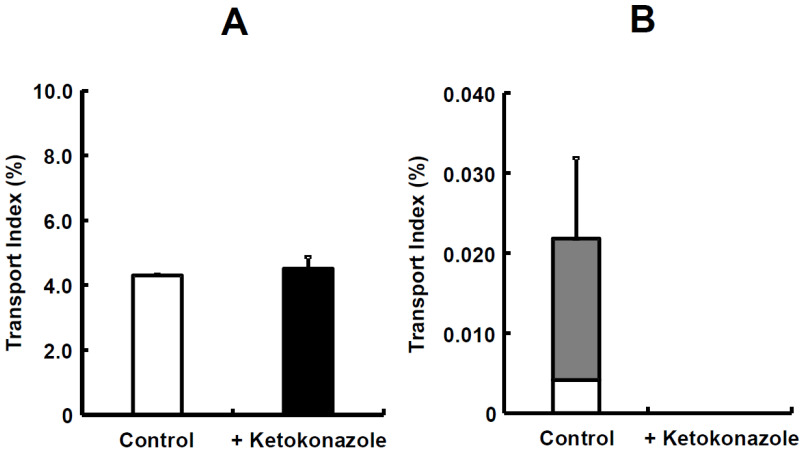
The inhibitory effect of ketoconazole on absorption of midazolam (**A**) and its metabolites (**B**) in dog small intestinal tissues. Each value of TI is the mean with SE of three experiments. Open column denotes 1-OH; closed column denotes 4-OH. Figure received with permission from [33].

**Figure 8 pharmaceutics-15-02732-f008:**
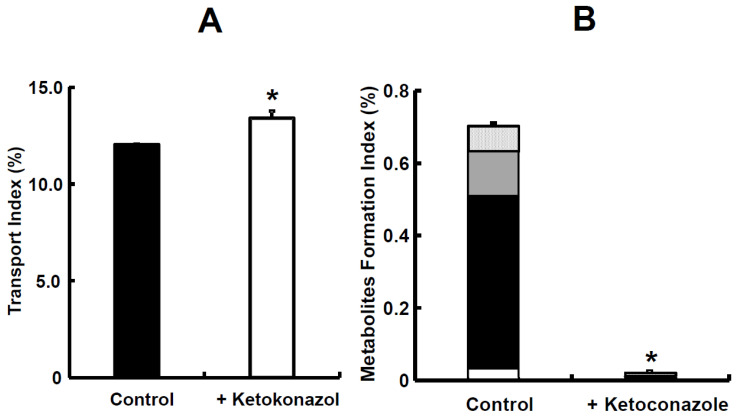
The inhibitory effect of ketoconazole on absorption of midazolam (**A**) and its metabolites (**B**) in rat small intestinal tissues. TI was calculated by [29]. MFI was calculated by [34]. Each value of TI and MFI is the mean with SE of three experiments. * *p* < 0.05 compared with data in the absence of ketoconazole. In (**B**), the open column denotes for 1-OH in apical; closed column denotes 1-OH in tissue; gray column denotes 1-OH in basal; patched column denotes 4-OH in tissue. Figure re-ceived with permission from [34].

**Figure 9 pharmaceutics-15-02732-f009:**
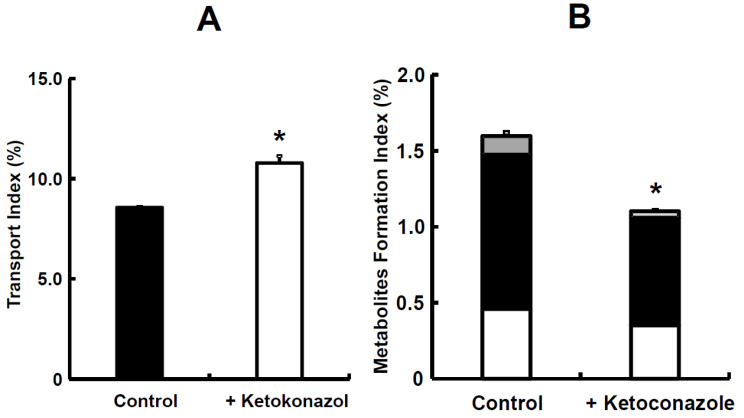
The inhibitory effect of ketoconazole on absorption of nifedipine (**A**) and its metabolite (**B**) in rat small intestinal tissues. TI was calculated by [29]. MFI was calculated by [34]. Each value of TI and MFI is the mean with SE of three experiments. * *p* < 0.05 compared with data in the absence of ketoconazole. In (**B**), the open column denotes oxidized nifedipine in apical; closed column denotes oxidized nifedipine in tissue; gray column denotes oxidized nifedipine in basal. Figure received with permission from [34].

**Table 1 pharmaceutics-15-02732-t001:** Gastric pH in young adults, adults, and older adults.

Population (Age Range, Number of Participants)	pH [Median (Range)]		Reference
	Fasted Condition ^1^	Fed Condition ^2^	
Young adults (21–35 years; *n* = 24)	1.7 (1.4–2.1)	5.0 (4.3–5.4)	[40]
Young adults (20–39 years; *n* = 5)	1.67 (1.24–5.95)	---	[39]
Adults (40–59 years; *n* = 6)	1.79 (1.29–4.52)	---	[39]
Older adults (60–79 years; *n* = 8)	2.12 (1.12–4.26)	---	[39]
Older adults (≧65 years; *n* = 79)	1.3 (1.1–1.6)	4.9 (3.9–5.5)	[41]

^1^ After an overnight fast or overnight fast and a glass of water. ^2^ After administration of a high-calorie, high-fat meal (800–1000 kcal; about 50% of calories are derived from fat). Table modified with permission from [38].

**Table 2 pharmaceutics-15-02732-t002:** pH in the lumen of duodenum, distal ileum and proximal colon of young adults and older adults.

Intestinal Segment	Population (Age Range, Number of Patients)	pH [Median (Range)]		Reference
		Fasted Condition ^1^	Fed Condition ^2^	
Duodenum	Young adults (21–35 years; *n* = 24) Young adults (18–25 years; *n* = 8)	6.1 (5.8–6.5) 6.84 (5.41–7.13)	6.3 (6.0–6.7)	[40,44]
	Older adults (65–83 years; *n* = 75) Older adults (62–72 years; *n* = 7)	6.5 * (6.2–6.7) 6.71 (6.50–7.28)	6.5 * (6.4–6.7)	[41,44]
Distal ileum	Young adults (22–42 years; *n* = 12)	8.0 (7.0–8.7)	8.1 (7.3–8.4)	[46]
	Older adults (65–70 years; *n* = 6 fasted, *n* = 4 fed)	7.9 (7.2–8.6)	6.7 * (6.1–7.9)	[45]
Proximal colon	Young adults (19–28 years; *n* = 12)	7.8 (6.4–8.4)	6.0 (5.3–7.9)	[47]
	Older adults (65–70 years; *n* = 8 fasted, *n* = 7 fed)	6.4 ** (6.1–7.6)	5.8 (5.7–6.4)	[45]

^1^ After an overnight fast or overnight fast and a glass of water. ^2^ After administration of a high-calorie, high-fat meal (800–1000 kcal; about 50% of calories are derived from fat). * Significantly different from data collected in young adults using the same methodology (Heidelberg capsule, [40]). ** Significantly different from data collected in young adults. Table modified with permission from [38].

**Table 3 pharmaceutics-15-02732-t003:** Buffer capacity of fluids in the distal ileum and proximal colon of young and older adults. All data in the table refer to acidic titration.

Intestinal Segment	Age group (Age Range, Number of Patients)	Buffer Capacity [mmol/L/ΔpH]		Reference
		Fasted Condition ^1^	Fed Condition ^2^	
Distal ileum	Young adults (22–42 years; *n* = 6 fasted, *n* = 7 fed)	8.9 ± 3.6	15.2 ± 8.4	[46]
	Older adults (65–70 years; *n* = 2 fed)	No data	20 and 35 *	[45]
Proximal colon	Young adults (19–28 years; *n* = 12)	21.5 ± 7.9	38.0 ± 16.0	[47]
	Older adults (65–70 years; *n* = 5)	45.0 ± 17.0 **	56.0 ± 11.0	[45]

^1^ After an overnight fast or overnight fast and a glass of water. ^2^ After administration of a high-calorie, high-fat meal (800–1000 kcal; about 50% of calories are derived from fat). * Individual data. ** Significantly different from young adults. Table modified with permission from [38].

**Table 4 pharmaceutics-15-02732-t004:** Osmolality of contents of the duodenum, distal ileum and proximal colon of young adults and older adults.

Intestinal Segment	Age Group (Age Range, Number of Subjects)	Osmolarity [mOsm/kg]		Reference
		Fasted Condition ^1^	Fed Condition ^2^	
Duodenum	Young adults (18–25 years; *n* = 8)	226 ± 35	---	[44]
	Older adults (62–72 years; *n* = 7)	215 ± 37	---	[44]
Distal ileum	Young adults (22–42 years; *n* = 12)	60 ± 50	252 ± 245	[46]
	Older adults (65–70 years; *n* = 5 fasted, *n* = 4 fed)	128 ± 56 *	193 ± 68	[45]
Proximal colon	Young adults (19–28 years; *n* = 11)	81 ± 102	224 ± 125	[47]
	Older adults (65–70 years; *n* = 4 fasted, *n* = 7 fed)	299 ± 49 *	264 ± 76	[45]

^1^ After an overnight fast or overnight fast and a glass of water. ^2^ After administration of a high-calorie, high-fat meal (800–1000 kcal; about 50% of calories are derived from fat). * Significantly different from young adults. Table modified with permission from [38].

**Table 5 pharmaceutics-15-02732-t005:** Protein levels of CYP enzymes and the major drug transporting ABC and SLC proteins in human jejunum.

	Jejunum	
	Median (Min–Max)	*n*
CYP3A4	13.4 (5–25.3)	37
CYP27A1	5.9 (4–10.2)	37
CYP4F2	5.4 (1.6–15.5)	37
CYP2S1	4.5 (1.6–7.5)	37
CYP2C9	1.5 (0.4–23.5)	37
CYP3A5	0.9 (0.1–3)	37
CYP20A1	0.8 (0.5–1.4)	37
CYP2C18	0.5 (0.2–6)	35
CYP4F12	0.5 (0.1–3.8)	37
CYP2D6; CYP2D7	0.4 (0.1–1.7)	32
CYP2J2	0.3 (0.1–0.9)	37
CYP1A1	0.2 (0–3.1)	35
CYP2C19	0.2 (0–0.6)	34
TAP2 (ABCB3)	2.2 (0.7–4.3)	37
TAP1 (ABCB2)	1.5 (0.8–3.4)	37
ABCB1	0.9 (0.3–1.9)	37
ABCG2	0.6 (0.3–1.5)	37
ABCC3	0.3 (0.1–0.5)	37
ABCC2	0.1 (0–0.2)	37
ABCC4	0.04 (0.003–0.1)	37
SLC15A1	0.9 (0.4–1.5)	37
SLC16A1	0.8 (0.3–1.7)	37
SLC19A3	0.04 (0.02–0.1)	13
SLC29A1	0.04 (0.02–0.1)	17
SLC51A	0.2 (0.03–0.6)	37
SLC51B	0.3 (0.2–0.6)	21
SLCO2B1	0.02 (0.01–0.2)	14

Protein levels of CYP enzymes and the major drug transporting ABC and SLC proteins in human jejunum biopsies from the 37 donors. Protein levels are given in fmol/μg protein. For proteins separated with “;” the specific isoforms could not be distinguished by the MaxQuant search engine due to large sequence overlap. max, maximum; min, minimum; *n*, number of donors for which the protein was quantified. Table modified with permission from [54].

## Data Availability

Data is contained within the article.

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
