# Peer review of "Simultaneous Prediction Method for Intestinal Absorption and Metabolism Using the Mini-Ussing Chamber System"

_pharmaceutics, 2023, doi:10.3390/pharmaceutics15122732_

Round 1

Reviewer 1 Report (Previous Reviewer 1)

Comments and Suggestions for Authors

The reviewers have considered the suggested comments, so the article is accepted in its current form.

Comments on the Quality of English Language

Minor editing of English language required

Author Response

Thanks for your comments.

Reviewer 2 Report (Previous Reviewer 3)

Comments and Suggestions for Authors

The authors have improved the manuscript from the previous submission. I have two comments:

1-The authors should review the title of the manuscript. The word "novel" may makes the reader think that it is a research paper, instead of a Review.

2- The authors may include a table with the data of intestinal absorption and metabolism that are available in the literature that used the mini-Ussing chamber system.   

Comments on the Quality of English Language

English should be revised, mainly in the information that was included in the manuscript and was highlighted in yellow color.

Author Response

Reviewer #2 (Remarks to the Author):

We thank Reviewer #2 for helpful comments on our manuscript.  We have revised the manuscript as outlined in the Response section below.

Reviewer 2

  • The authors should review the title of the manuscript. The word "novel" may makes the reader think that it is a research paper, instead of a Review.

In order to response to the Reviewer’s comment, we changed the title. 

2- The authors may include a table with the data of intestinal absorption and metabolism that are available in the literature that used the mini-Ussing chamber system.

We thank the Reviewer for this suggestion.  But the data and literature regarding metabolism that used by Ussing chamber system was nothing. 

Reviewer 3 Report (Previous Reviewer 4)

Comments and Suggestions for Authors

The manuscript entitled “Novel simultaneous prediction method of intestinal absorption and metabolism using the mini-Ussing chamber system” describes the advantage of application of Ussig-chamber in the simultaneous prediction of intestinal absorption and metabolism. The article is principally well-written, contains useful information about this ex vivo system supported by literary references, moreover, the authors significantly improved the quality of manuscript in comparison to previous version, however, the authors still did not use the template of the journal! Besides that, with the new submission the the authors adressed all my concerns.

Author Response

Thanks for your comments.

Reviewer 4 Report (New Reviewer)

Comments and Suggestions for Authors

The current review manuscript discussing the predictive models employed for investigating intestinal absorption and metabolism is exciting and novel. All the comments are well addressed by the authors. The revised version of the manuscript can be accepted for publication.

Comments on the Quality of English Language

Minor English editing is needed 

Author Response

Thanks for your comments.

This manuscript is a resubmission of an earlier submission. The following is a list of the peer review reports and author responses from that submission.

Round 1

Reviewer 1 Report

Comments and Suggestions for Authors

It is an article that addresses an interesting topic that is the methods to predict intestinal absorption of drugs and metabolism. There are points that need to be considered by the authors:

- In the Introducction section, page 2, line 49. Please add information justifying that it is the most convenient route of administration to have a better adherence of patients to treatment.

- In section 2.2, it would be highly recommended to include a table that condenses in a general way all the information described in the text, since it is relevant and it is a way of making it more accessible to readers.

- In section 3, like the comment suggested above, generate a table summarizing everything related to intestinal transporters and enzymes.

- In section 4, Generate a table from this section. It might even be quite nice to include an overview table that encompasses everything discussed in the previous sections. I think it would be very useful for readers to present it that way.

In general, the information presented is very interesting, but I think that it would have to be restructured and a general analysis of everything described would be needed to close the topic properly.

Comments on the Quality of English Language

Minor editing of English language required.

Reviewer 2 Report

Comments and Suggestions for Authors

The authors prepared a review manuscript for simultaneous prediction of intestinal absorption and metabolism using the mini-ussing chamber system. 

Please see the comments below and provide the responses and modify the manuscript accordingly.

1. Is this a review or research manuscript? I got confused the manuscript has research figures but there is no description about materials, methods, equations and approach.

2.  The authors need to provide concept diagram and summary figure for simultaneous prediction of intestinal absorption and metabolism. 

3. How the figures were prepared? The authors mentioned only one source of reference. This is a review article with collection of several articles. The authors can include the figures from multiple sources to strengthen the manuscript.

4. The authors can include the physiology of gastrointestinal tract in section 2.

5. The authors directly introduced section 4 and 5 with TI and MFI without providing the sufficient background information, methods, equations, etc.

6. The conclusion should be expanded with importance of this approach.

Comments on the Quality of English Language

There are several spelling mistakes in the axis legends of the figures and in the manuscript content. The authors need to check and correct the spelling mistakes throughout the manuscript.

Reviewer 3 Report

Comments and Suggestions for Authors

The manuscript presented by Kondo & Miyake review the literature regarding the use of mini-Ussing chamber for the prediction of intestinal absorption of drugs.

The manuscript is well written and correct. However, it is very simple, without an in dept discussion. The manuscript could be submitted to a less demanding journal.

Reviewer 4 Report

Comments and Suggestions for Authors

The manuscript entitled “Novel simultaneous prediction method of intestinal absorption and metabolism using the mini-Ussing chamber system” describes the advantage of application of Ussig-chamber in the simultaneous prediction of intestinal absorption and metabolism. The article is principally well-written, contains useful information about this ex vivo system supported by literary references, however some minor concerns raised in the reviewer, which can improve the quality of manuscript!

The Figures are missing from manuscript, please embed them also in the template!

A table collecting the pH values of different GI states (Section 2.2) would be useful for the reader!